# ON THE EVALUATION OF CONDITIONAL GANS

## ABSTRACT

Conditional Generative Adversarial Networks (cGANs) are finding increasingly widespread use in many application domains. Despite outstanding progress, quantitative evaluation of such models often involves multiple distinct metrics to assess different desirable properties, such as image quality, conditional consistency, and intra-conditioning diversity. In this setting, model benchmarking becomes a challenge, as each metric may indicate a different "best" model. In this paper, we propose the Fréchet Joint Distance (FJD), which is defined as the Fréchet distance between *joint distributions* of images and conditioning, allowing it to implicitly capture the aforementioned properties in a *single metric*. We conduct proof-of-concept experiments on a controllable synthetic dataset, which consistently highlight the benefits of FJD when compared to currently established metrics. Moreover, we use the newly introduced metric to compare existing cGAN-based models for a variety of conditioning modalities (e.g. class labels, object masks, bounding boxes, images, and text captions). We show that FJD can be used as a promising *single* metric for cGAN benchmarking and model selection.

## 1 INTRODUCTION

The use of generative models is growing across many domains (van den Oord et al., 2016c; Vondrick et al., 2016; Serban et al., 2017; Karras et al., 2018; Brock et al., 2019). Among the most promising approaches, Variational Auto-Encoders (VAEs) (Kingma & Welling, 2014), auto-regressive models (van den Oord et al., 2016a;b), and Generative Adversarial Networks (GANs) (Goodfellow et al., 2014) have been driving significant progress, with the latter at the forefront of a wide-range of applications (Mirza & Osindero, 2014; Reed et al., 2016; Zhang et al., 2018a; Vondrick et al., 2016; Almahairi et al., 2018; Subramanian et al., 2018; Salvador et al., 2019). In particular, significant research has emerged from practical applications, which require generation to be based on existing context. For example, tasks such as image inpainting, super-resolution, or text-to-image synthesis have been successfully addressed within the framework of *conditional* generation, with conditional GANs (cGANs) among the most competitive approaches. Despite these outstanding advances, quantitative evaluation of GANs remains a challenge (Theis et al., 2016; Borji, 2018).

In the last few years, a significant number of evaluation metrics for GANs have been introduced in the literature (Salimans et al., 2016; Heusel et al., 2017; Bińkowski et al., 2018; Shmelkov et al., 2018; Zhou et al., 2019; Kynkäänniemi et al., 2019; Ravuri & Vinyals, 2019). Although there is no clear consensus on which quantitative metric is most appropriate to benchmark GAN-based models, Inception Score (IS) (Salimans et al., 2016) and Fréchet Inception Distance (FID) (Heusel et al., 2017) have been extensively used. However, both IS and FID were introduced in the context of *unconditional* image generation and, hence, focus on capturing certain desirable properties such as *visual quality* and *sample diversity*, which do not fully encapsulate all the different phenomena that arise during conditional image generation.

In *conditional* generation, we care about *visual quality*, *conditional consistency* – i.e., verifying that the generation respects its conditioning, and *intra-conditioning diversity* – i.e., sample diversity per conditioning. Although visual quality is captured by both metrics, IS is agnostic to intra-conditioning diversity and FID only captures it indirectly.[1] Moreover, neither of them can capture conditional con-

---

[1]FID compares image distributions and, as such, should be able to roughly capture the intra-conditioning diversity. Since it cares about the image marginal distribution exclusively, it fails to capture intra-conditioning diversity when changes only affect the image-conditioning joint distribution. See Appendix A.

sistency. In order to overcome these shortcomings, researchers have resorted to reporting conditional consistency and diversity metrics in conjunction with FID (Zhao et al., 2019; Park et al., 2019).

Consistency metrics often use some form of concept detector to ensure that the requested conditioning appears in the generated image as expected. Although intuitive to use, these metrics require pre-trained models that cover the same target concepts in the same format as the conditioning (i.e., classifiers for image-level class conditioning, semantic segmentation for mask conditioning, etc.), which may or may not be available off-the-shelf. Moreover, using different metrics to evaluate different desirable properties may hinder the process of model selection, as there may not be a single model that surpasses the rest in all measures. In fact, it has recently been demonstrated that there is a natural trade-off between image quality and sample diversity (Yang et al., 2019), which calls into question how we might select the correct balance of these properties.

In this paper we introduce a new metric called Fréchet Joint Distance (FJD), which is able to implicitly assess image quality, conditional consistency, and intra-conditioning diversity. FJD computes the Fréchet distance on an embedding of the joint image-conditioning distribution, and introduces only small computational overhead over FID compared to alternative methods. We evaluate the properties of FJD on a variant of the synthetic dSprite dataset (Matthey et al., 2017) and verify that it successfully captures the desired properties. We provide an analysis on the behavior of both FID and FJD under different types of conditioning such as class labels, bounding boxes, and object masks, and evaluate a variety of existing cGAN models for real-world datasets with the newly introduced metric. Our experiments show that (1) FJD captures the three highlighted properties of conditional generation; (2) it can be applied to any kind of conditioning (e.g., class, bounding box, mask, image, text, etc.); and (3) when applied to existing cGAN-based models, FJD demonstrates its potential to be used as a promising *unified* metric for hyper-parameter selection and cGAN benchmarking. To our knowledge, there are no existing metrics for conditional generation that capture all of these key properties.

## 2 RELATED WORK

Conditional GANs have witnessed outstanding progress in recent years. Training stability has been improved through the introduction of techniques such as progressive growing, Karras et al. (2018), spectral normalization (Miyato et al., 2018) and the two time-scale update rule (Heusel et al., 2017). Architecturally, conditional generation has been improved through the use of auxiliary classifiers (Odena et al., 2017) and the introduction of projection-based conditioning for the discriminator (Miyato & Koyama, 2018). Image quality has also benefited from the incorporation of self-attention (Zhang et al., 2018a), as well as increases in model capacity and batch size (Brock et al., 2019).

All of this progress has led to impressive results, paving the road towards the challenging task of generating more complex scenes. To this end, a flurry of works have tackled different forms of conditional image generation, including class-based (Mirza & Osindero, 2014; Heusel et al., 2017; Miyato et al., 2018; Odena et al., 2017; Miyato & Koyama, 2018; Brock et al., 2019), image-based (Isola et al., 2017; Zhu et al., 2017a; Wang et al., 2018; Zhu et al., 2017b; Almahairi et al., 2018; Huang et al., 2018; Mao et al., 2019), mask- and bounding box-based (Hong et al., 2018; Hinz et al., 2019; Park et al., 2019; Zhao et al., 2019), as well as text- (Reed et al., 2016; Zhang et al., 2017; 2018a; Xu et al., 2018; Hong et al., 2018) and dialogue-based conditionings (Sharma et al., 2018; El-Nouby et al., 2019). This intensified research has lead to the development of a variety of metrics to assess the three factors of conditional image generation process quality, namely: visual quality, conditional consistency, and intra-conditioning diversity.

**Visual quality**. A number of GAN evaluation metrics have emerged in the literature to assess visual quality of generated images in the case of unconditional image generation. Most of these metrics either focus on the separability between generated images and real images (Lehmann & Romano, 2005; Radford et al., 2016; Yang et al., 2017; Isola et al., 2017; Zhou et al., 2019), compute the distance between distributions (Gretton et al., 2012; Heusel et al., 2017; Arjovsky et al., 2017), assess sample quality and diversity from conditional or marginal distributions (Salimans et al., 2016; Gurumurthy et al., 2017; Zhou et al., 2018), measure the similarity between generated and real images (Wang et al., 2004; Xiang & Li, 2017; Snell et al., 2017; Juefei-Xu et al., 2017) or are log-likelihood based (Theis et al., 2016)[2]. Among these, the most accepted automated visual quality metrics are

---

[2]We refer the reader to (Borji, 2018) for a detailed overview and insightful discussion of existing metrics.

Inception Score (IS) (Salimans et al., 2016) and Fréchet Inception Distance (FID) (Heusel et al., 2017).

**Conditional consistency**. To assess the consistency of the generated images with respect to model conditioning, researchers have reverted to available, pre-trained feed-forward models. The structure of these models depends on the modality of the conditioning (e.g. segmentation models are used for mask conditioning or image captioning models are applied to evaluate text conditioning). Moreover, the metric used to evaluate the forward model on the generated distribution depends on the conditioning modality and includes: accuracy in the case of class-conditioned generation, Intersection over Union when using bounding box- and mask-conditionings, BLEU (Papineni et al., 2002), METEOR (Banerjee & Lavie, 2005) or CIDEr (Vedantam et al., 2015) in the case of text-based conditionings, and Structural Similarity (SSIM) or peak signal-to-noise ratio (PSNR) for image-conditioning.

**Intra-conditioning diversity**. The most common metric for evaluating sample diversity is Learned Perceptual Image Patch Similarity (LPIPS) (Zhang et al., 2018b), which measures the distance between samples in a learned feature space. Alternatively, (Miyato & Koyama, 2018) proposed Intra-FID, which calculates a FID score separately for each conditioning and reports the average score over all conditionings. This method should in principle capture the desirable properties of image quality, conditional consistency, and intra-class diversity. However, it scales poorly with the number of unique conditions, as the computationally intensive FID calculation must be repeated for each case, and because FID behaves poorly when the sample size is small (Bińkowski et al., 2018). Furthermore, in cases where the conditioning cannot be broken down into a set of discrete classes (e.g., pixel-based conditioning), Intra-FID is intractable. As a result, it has not been applied beyond class-conditioning.

## 3 REVIEW OF FRÉCHET INCEPTION DISTANCE (FID)

FID aims to compare the statistics of generated samples to samples from a real dataset. Given two multivariate Gaussian distributions $\mathcal{N}(\boldsymbol{\mu}, \boldsymbol{\Sigma})$ and $\mathcal{N}(\hat{\boldsymbol{\mu}}, \hat{\boldsymbol{\Sigma}})$, Fréchet Distance (FD) is defined as:

$$d^2\left((\boldsymbol{\mu}, \boldsymbol{\Sigma}), (\hat{\boldsymbol{\mu}}, \hat{\boldsymbol{\Sigma}})\right) = ||\boldsymbol{\mu} - \hat{\boldsymbol{\mu}}||_2^2 + Tr\left(\boldsymbol{\Sigma} + \hat{\boldsymbol{\Sigma}} - 2(\boldsymbol{\Sigma}\hat{\boldsymbol{\Sigma}})^{1/2}\right). \tag{1}$$

When evaluating a generative model, $\mathcal{N}(\boldsymbol{\mu}, \boldsymbol{\Sigma})$ represents the data (reference) distribution, obtained by fitting a Gaussian to samples from a reference dataset, and $\mathcal{N}(\hat{\boldsymbol{\mu}}, \hat{\boldsymbol{\Sigma}})$ represents the learned (generated) distribution, a result of fitting to samples from a generative model.

In FID, both the real images and model samples are embedded in a learned feature space using a pre-trained Inception v3 model (Szegedy et al., 2016). Thus, the Gaussian distributions are defined in the embedded space. More precisely, given a dataset of images $\{\mathbf{x}^{(i)}\}_{i=0}^{N}$, a set of model samples $\{\hat{\mathbf{x}}^{(i)}\}_{i=0}^{M}$, and an Inception embedding function $f$, we estimate the Gaussian parameters $\boldsymbol{\mu}$, $\boldsymbol{\Sigma}$, $\hat{\boldsymbol{\mu}}$ and $\hat{\boldsymbol{\Sigma}}$ as:

$$\boldsymbol{\mu} = \frac{1}{N}\sum_{i=0}^{N} f(\mathbf{x}^{(i)}), \quad \boldsymbol{\Sigma} = \frac{1}{N-1}\sum_{i=0}^{N}\left(f(\mathbf{x}^{(i)}) - \boldsymbol{\mu}\right)\left(f(\mathbf{x}^{(i)}) - \boldsymbol{\mu}\right)^T, \tag{2}$$

$$\hat{\boldsymbol{\mu}} = \frac{1}{M}\sum_{i=0}^{M} f(\hat{\mathbf{x}}^{(i)}), \quad \hat{\boldsymbol{\Sigma}} = \frac{1}{M-1}\sum_{i=0}^{M}\left(f(\hat{\mathbf{x}}^{(i)}) - \hat{\boldsymbol{\mu}}\right)\left(f(\hat{\mathbf{x}}^{(i)}) - \hat{\boldsymbol{\mu}}\right)^T. \tag{3}$$

## 4 FRÉCHET JOINT DISTANCE (FJD)

In conditional image generation, a dataset is composed of image-condition pairs $\{(\mathbf{x}^{(i)}, \mathbf{y}^{(i)})\}_{i=0}^{N}$, where the conditioning can take variable forms, such as image-level classes, segmentation masks, or text. The goal of conditional image generation is to produce realistic looking, diverse images $\hat{\mathbf{x}}$ that are *consistent* with the conditioning $\hat{\mathbf{y}}$. Thus, a set of model samples with corresponding conditioning can be defined as: $\{(\hat{\mathbf{x}}^{(i)}, \hat{\mathbf{y}}^{(i)})\}_{i=0}^{M}$.

As discussed in Section 3, the Fréchet distance (FD) compares any two Gaussians defined over arbitrary spaces. In FJD, we propose to compute the FD between two Gaussians defined over the *joint image-conditioning embedding space*.

More precisely, given an image embedding function $f$, a conditioning embedding function $h$, a conditioning embedding scaling factor $\alpha$, and a merging function $g$ that combines the image embedding with the conditioning embedding into a joint one, we can estimate the respective Gaussian parameters $\boldsymbol{\mu}$, $\boldsymbol{\Sigma}$, $\hat{\boldsymbol{\mu}}$ and $\hat{\boldsymbol{\Sigma}}$ as:

$$\boldsymbol{\mu} = \frac{1}{N} \sum_{i=0}^{N} g\left(f(\mathbf{x}^{(i)}), \alpha h(\mathbf{y}^{(i)})\right), \quad \hat{\boldsymbol{\mu}} = \frac{1}{M} \sum_{i=0}^{M} g\left(f(\hat{\mathbf{x}}^{(i)}), \alpha h(\hat{\mathbf{y}}^{(i)})\right), \quad (4)$$

$$\boldsymbol{\Sigma} = \frac{1}{N-1} \sum_{i=0}^{N} \left(g\left(f(\mathbf{x}^{(i)}), \alpha h(\mathbf{y}^{(i)})\right) - \boldsymbol{\mu}\right)\left(g\left(f(\mathbf{x}^{(i)}), \alpha h(\mathbf{y}^{(i)})\right) - \boldsymbol{\mu}\right)^T, \quad (5)$$

$$\hat{\boldsymbol{\Sigma}} = \frac{1}{M-1} \sum_{i=0}^{M} \left(g\left(f(\hat{\mathbf{x}}^{(i)}), \alpha h(\hat{\mathbf{y}}^{(i)})\right) - \hat{\boldsymbol{\mu}}\right)\left(g\left(f(\hat{\mathbf{x}}^{(i)}), \alpha h(\hat{\mathbf{y}}^{(i)})\right) - \hat{\boldsymbol{\mu}}\right)^T. \quad (6)$$

Note that by computing the FD over the joint image-conditioning distribution, we are able to simultaneously assess image quality, conditional consistency, and intra-conditioning diversity, all of which are important factors in evaluating the quality of conditional image generation models.

To ensure reproducibility, when reporting FJD scores it is important to include details such as which conditioning embedding function was used, which dataset is used for the reference distribution, and the $\alpha$ value. We report these values for all of our experiments in Appendix B.

### 4.1    CONDITIONING EMBEDDING FUNCTION: $h$

The purpose of the embedding function $h$ is to reduce the dimensionality and extract a useful feature representation of the conditioning. As such, the choice of $h$ will vary depending on the modality of conditioning. In most cases, an off-the-shelf, pretrained embedding can be used for the purposes of extracting a useful representation. In the absence of preexisting, pretrained conditioning

Table 1: Suggested embedding functions for different conditioning modalities.

| Conditioning Modality | Embedding Function |
|---|---|
| Class / Attribute labels | One-hot encoding |
| Bounding boxes / Masks | Regularized AE (Ghosh et al., 2019) |
| Images | Inceptionv3 (Szegedy et al., 2016) |
| Captions / Dialogue | Sentence-BERT (Reimers & Gurevych, 2019) |

embedding functions, a new one should be learned. For example, for bounding box and mask conditionings the embedding function could be learned with an autoencoder. [3] For suggested assignments of conditioning modalities to embedding functions please refer to Table 1.

### 4.2    CONDITIONING EMBEDDING SCALING FACTOR: $\alpha$

In order to control the relative contribution of the image component and the conditioning component to the final FJD value, we scale the conditioning embedding by a constant $\alpha$. In essence, $\alpha$ indicates how much we care about the conditioning component compared to the image component. When $\alpha = 0$, the conditioning component is ignored and FJD is equivalent to FID. As the value of $\alpha$ increases, the perceived importance of the conditioning component is also increased and reflected accordingly in the resulting measure. To equally weight the image component and the conditioning component, we recommend setting $\alpha$ to be equal to the ratio between the average $L_2$ norm of the image embedding and the conditioning embedding. This weighting ensures that FJD retains consistent behaviour across conditioning embeddings, even with varying dimensionality or magnitude. We note that $\alpha$ should be calculated on data from the reference distribution (real data distribution), and then applied to all conditioning embeddings thereafter. See Appendix F for an example of the effect of the $\alpha$ hyperparameter.

---

[3] In the initial stages of this project, we also explored methods to bypass this additional training step by projecting a visual representation of bounding box or mask conditioning into an Inceptionv3 embedding space. However, the Inceptionv3 embedding may not properly capture object positions as it is trained to classify, discarding precise spatial information. Therefore, we consider autoencoders (AE) to be better suited to our setup since they are trained to recover both object appearance and spatial information from the embedded representation.

### 4.3 MERGING FUNCTION: $g$

The purpose of the merging function $g$ is to combine the image embedding and conditioning embedding into a single joint embedding. We compared several candidate merging functions and found concatenation of the image embedding and conditioning embedding vectors to be most effective, both in terms of simplicity and performance. As such, concatenation is used as the merging function in all following experiments.

## 5 EVALUATION OF THE PROPERTIES OF FRÉCHET JOINT DISTANCE

In this section, we demonstrate that FJD captures the three desiderata of conditional image generation, namely image quality, conditional consistency and intra-conditioning diversity.

### 5.1 DATASET

**dSprite-textures.** The dSprite dataset (Matthey et al., 2017) is a synthetic dataset where each image depicts a simple 2D shape on a black background. Each image can be fully described by a set of factors, including shape, scale, rotation, $x$ position, and $y$ position. We augment the dSprite dataset to create *dSprite-textures* by adding three texture patterns for each sample. Additionally, we include class labels indicating shape, as well as bounding boxes and mask labels for each sample (see Figure 1). In total, the dataset contains 2,211,840 unique images. This synthetic dataset allows us to exactly control our sample distribution and, thereby, simulate a generator with image-conditioning inconsistencies or reduced sample diversity. To embed the conditioning for calculating FJD in the following experiments, we use one-hot encoding for the class labels, and autoencoder representations for the bounding box and mask labels.[4] We are releasing the code to generate dSprite-textures.

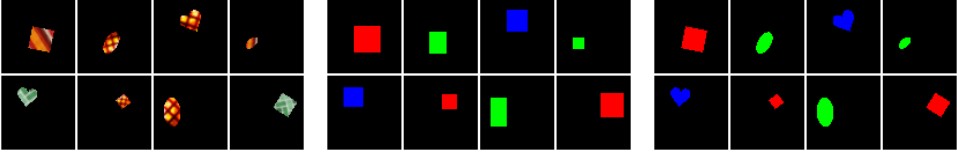

Figure 1: Left: dSprite-textures images. Center: Bounding box labels. Right: Mask labels.

### 5.2 IMAGE QUALITY

In this subsection, we aim to test the sensitivity of FJD to image quality perturbations. To do so, we draw 10k random samples from the dSprite-textures dataset to form a reference dataset. The generated dataset is simulated by duplicating the reference dataset and adding Gaussian noise drawn from $\mathcal{N}(0, \sigma)$ to the images, where $\sigma \in [0, 0.25]$ and pixel values are normalized (and clipped after noise addition) to the range $[0, 1]$. The addition of noise mimics a generative model that produces low quality images. We repeat this experiment for all three conditioning types in dSprite-textures: class, bounding box, and mask.

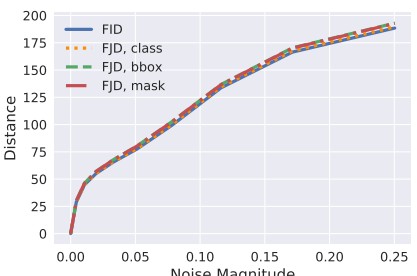

Figure 2: **Image quality:** FID and FJD exhibit similar trends for class, bounding box, and mask conditioning under varying noise levels added to images.

Results are shown in Figure 2, where we plot both FID and FJD as a function of the added Gaussian noise ($\sigma$ is indicated on the $x$-axis as Noise Magnitude). We find that, in all cases, FJD has a very similar trend to FID, indicating that it successfully captures image quality. Additional image quality experiments on the large scale COCO-Stuff dataset can be found in Appendix C.

---

[4]Architectural details for autoencoders used in this paper can be found in Appendix G.

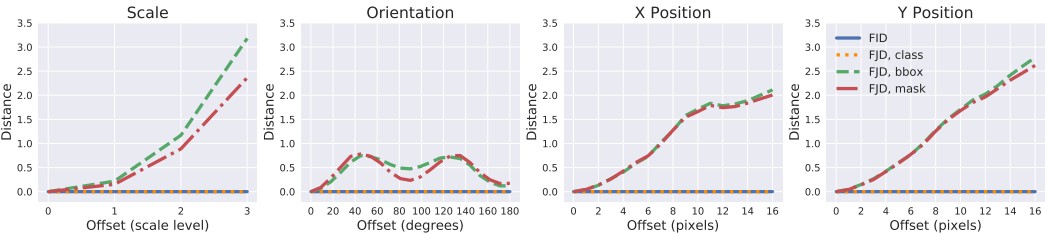

Figure 3: **Conditional consistency:** Change in FJD with respect to offset on Dsprite-textures dataset for class, bounding box and mask conditionings.

### 5.3 CONDITIONAL CONSISTENCY

In this subsection, we aim to highlight the sensitivity of FJD to conditional consistency. In particular, we target specific types of inconsistencies, such as incorrect scale, orientation, or position. We draw a set of 10k samples from the dSprite-textures dataset and duplicate it to represent the reference dataset and the generated dataset, each with identical image and conditioning marginal distributions. For 30% of the generated dataset samples we swap conditionings of pairs of samples that are identical in all but one of the attributes (scale, orientation, $x$ position or $y$ position). For example, if one generated sample has attribute $x$ position 4 and a second generated sample has attribute $x$ position 7, swapping their conditionings leads to generated samples that are offset by 3 pixels w.r.t. their ground truth $x$ position. Swapping conditionings in this manner allows us to control for specific attributes' conditional consistency, while keeping the image and conditioning marginal distributions unchanged. As a result, all changes in FJD can be attributed solely to conditional inconsistencies.

Figure 3 depicts the results of this experiment for four different types of alterations: scale, orientation, and $x$ and $y$ positions. We observe that the FID between image distributions (solid blue line) remains constant even as the degree of conditional inconsistency increases. For class conditioning (dotted orange line), FJD also remains constant, as changes to scale, orientation, and position are independent of the object class. Bounding box and mask conditionings, as they contain spatial information, produce variations in FJD that are proportional to the offset. Interestingly, for the orientation offsets, FJD with mask conditioning fluctuates rather than increasing monotonically. This behaviour is due to the orientation masks partially re-aligning with the ground truth around $90°$ and $180°$. Each of these cases emphasize the effective sensitivity of FJD with respect to conditional consistency. Additional conditional consistency experiments with text conditioning can be found in Appendix D.

### 5.4 INTRA-CONDITIONING DIVERSITY

In this subsection, we aim to test the sensitivity of FJD to intra-conditioning diversity[5], by alternating the per-conditioning image texture variability. More precisely, we vary the texture based on four different image attributes: *shape* that is captured in all tested conditionings, as well as *scale*, *orientation* and *position* that are captured by bounding box and mask conditionings only. To create attribute-texture assignments, we stratify attributes based on their values. For example, one possible shape-based stratification of a dataset with three shapes might be: [squares, ellipses, hearts]. To quantify the dataset intra-conditioning diversity, we introduce a diversity score. A diversity score of 1 means that the per-attribute texture distribution is uniform across stratas, while a diversity score of 0 means that each strata is assigned to a single texture. Middling diversity scores indicate that the textural distribution is skewed towards one texture type in each strata. We create our reference dataset by randomly drawing 10k samples. The generated distribution is created by duplicating the reference distribution and adjusting the per-attribute texture variability to achieve the desired diversity score.

The results of these experiments are shown in Figure 4, which plots the increase in FID and FJD, for different types of conditioning, as the diversity of textures within each subset decreases. For all tested scenarios, we observe that FJD is sensitive to intra-conditioning diversity changes. Moreover, not surprisingly, since a change in the joint distribution of attributes and textures also implies a change to the image marginal distribution, we observe that FID increases with reduced diversity. This

---

[5]Note that for real datasets, intra-conditioning diversity is most often reduced as the strength of conditioning increases (e.g., mask conditionings usually present a single image instantiation, presenting no diversity).

Figure 4: **Intra-conditioning diversity:** FJD and FID as intra-conditioning diversity decreases.

experiment suggests that FID is able to capture intra-conditioning diversity changes when the image conditional distribution is also affected. However, if the image marginal distribution were to stay constant, FID would be blind to intra-conditioning diversity changes (as is shown in Section 5.3).

## 6 EVALUATION OF EXISTING CONDITIONAL GENERATION MODELS

In this section, we seek to demonstrate the application of FJD to evaluate models with several different conditioning modalities, in contrast to FID and standard conditional consistency and diversity metrics. We focus on testing class-conditioned, image-conditioned, and text-conditioned image generation tasks, which have been the focus of numerous works[6]. Multi-label, bounding box, and mask conditioning are also explored in Appendix I. We note that FJD and FID yield similar rankings of models in this setting, which is to be expected since most models use similar conditioning mechanisms. Rankings are therefore dominated by image quality, rather than conditional consistency. We refer the reader to Appendix F and H for examples of cases where FJD ranks models differently than FID.

**Class-conditioned cGANs.** Table 2 compares three state-of-the-art class-conditioned generative models trained on ImageNet at $128 \times 128$ resolution. Specifically, we evaluate SN-GAN (Miyato et al., 2018) trained with and without a projection discriminator (Miyato & Koyama, 2018), and BigGAN (Brock et al., 2019). Accuracy is used to evaluate conditional consistency, and is com-

Table 2: Comparison of class-conditioned models trained on ImageNet (resolution $128 \times 128$).

|                      | FJD ↓ | FID ↓ | Acc. ↑ | Diversity ↑ |
|----------------------|-------|-------|--------|-------------|
| **SN-GAN (concat)**  | 63.7  | 39.8  | 18.2   | **0.622**   |
| **SN-GAN (proj)**    | 41.7  | 27.4  | 35.7   | 0.612       |
| **BigGAN**           | **17.0** | **9.55** | **67.4** | 0.550    |

puted as the Inception v3 accuracy of each model's generated samples, using their conditioning as classification ground truth. Class labels from the validation set are used as conditioning to generate 50k samples for each model, and the training set is used as the reference distribution. One-hot encoding is used to embed the class conditioning for the purposes of calculating FJD.

We find that FJD follows the same trend as FID for class-conditioned models, preserving their ranking and highlighting the FJD's ability to capture image quality. Additionally, we note that the difference between FJD and FID correlates with each model's classification accuracy, with smaller gaps appearing to indicate better conditional consistency. Diversity scores, however, rank models in the opposite order compared to all other metrics.

This behaviour evokes the trade-off between realism and diversity highlighted by Yang et al. (2019). Ideally, we would like a model that produces diverse outputs, but this property is not as attractive if it also results in a decrease in image quality. At what point should diversity be prioritized over image quality, and vice versa? FJD is a suitable metric for answering this question if the goal is to find a model that best matches the target conditional data generating distribution.

**Image-conditioned cGANs.** Table 3 compares four state-of-the-art image translation models: Pix2pix (Isola et al., 2017), BicycleGAN (Zhu et al., 2017b), MSGAN (Mao et al., 2019), and MUNIT (Huang et al., 2018). We evaluate on four different image-to-image datasets: Facades (Tyleček & Šára, 2013), Maps (Isola et al., 2017), Edges2Shoes and Edges2Handbag (Zhu et al.,

---

[6]A list of pre-trained models used in these evaluations can be found in Appendix E.

Table 3: Comparison of image-conditioned models. Results averaged over 5 runs.

| Dataset | Facades | | | | Maps | | | |
|---|---|---|---|---|---|---|---|---|
| | FJD ↓ | FID ↓ | Consistency ↓ | Diversity ↑ | FJD ↓ | FID ↓ | Consistency ↓ | Diversity ↑ |
| **Pix2pix** | 161.3 | 104.0 | **0.413** | 0.056 | 233.4 | 106.8 | **0.444** | 0.049 |
| **BicycleGAN** | **145.9** | **85.0** | 0.436 | 0.289 | **220.4** | **93.2** | 0.449 | 0.247 |
| **MSGAN** | 152.4 | 93.1 | 0.478 | **0.376** | 249.3 | 123.3 | 0.478 | **0.452** |
| | Edges2Shoes | | | | Edges2Handbags | | | |
| **Pix2pix** | 115.4 | 74.2 | **0.215** | 0.040 | 162.3 | 95.6 | **0.314** | 0.042 |
| **BicycleGAN** | **88.2** | **47.3** | 0.239 | 0.191 | **142.1** | **76.0** | 0.324 | 0.252 |
| **MUNIT** | 98.1 | 56.2 | 0.270 | **0.229** | 147.9 | 79.1 | 0.382 | **0.339** |

2016). To assess conditional consistency we utilize LPIPS to measure the average distance between generated images and their corresponding ground truth images. Conditioning from the validation sets are used to generate images, while the training sets are used as reference distributions. An Inceptionv3 model is used to embed the image conditioning for the FJD calculation. Due to the small size of the validation sets, we report scores averaged over 5 evaluations of each model.

In this setting we encounter some ambiguity with regards to model selection, as for all datasets, each metric ranks the models differently. BicycleGAN appears to have the best image quality, Pix2pix produces images that are most visually similar to the ground truth, and MSGAN and MUNIT achieve the best sample diversity scores. This scenario demonstrates the benefits of using a single unified metric for model selection, for which there is only a single best model.

**Text-conditioned cGANs.** Table 4 shows FJD and FID scores for three state-of-the-art text-conditioned models trained on the Caltech-UCSD Birds 200 dataset (CUB-200) (Welinder et al., 2010) at $256 \times 256$ resolution: HDGan (Zhang et al., 2018c), StackGAN++ (Zhang et al., 2018a), and AttnGAN (Xu et al., 2018). Conditional consistency is evaluated us-

Table 4: Comparison of text-conditioned models trained on CUB-200 (resolution $256 \times 256$).

| | FJD ↓ | FID ↓ | VS sim. ↑ | Diversity ↑ |
|---|---|---|---|---|
| **HDGan** | 26.1 | 23.3 | 0.340 | **0.687** |
| **StackGAN++** | 21.8 | 18.4 | 0.341 | 0.652 |
| **AttnGAN** | **16.7** | **13.6** | **0.477** | 0.625 |

ing visual-semantic similarity, as proposed by Zhang et al. (2018c). Conditioning from the test set captions is used to generate 30k images, and the same test set is also used as the reference distribution. We use pre-computed Char-CNN-RNN sentence embeddings as the conditioning embedding for FJD, since they are commonly used with CUB-200 and are readily available.

In this case we find that AttnGAN dominates in terms of conditional consistency compared to HDGan and StackGAN++, while all models are comparable in terms of diversity. AttnGAN is ranked best overall by FJD. In cases where the biggest differentiator between the models is image quality, FID and FJD will provide a consistent ranking as we see here. In cases where the trade-off is more subtle we believe practitioners will opt for a metric that measurably captures intra-conditioning diversity.

## 7 CONCLUSIONS

In this paper we introduce Fréchet Joint Distance (FJD), which is able to assess image quality, conditional consistency, and intra-conditioning diversity within a single metric. We compare FJD to FID on the synthetic dSprite-textures dataset, validating its ability to capture the three properties of interest across different types of conditioning, and highlighting its potential to be adopted as a unified cGAN benchmarking metric. We also demonstrate how FJD can be used to address the potentially ambiguous trade-off between image quality and sample diversity when performing model selection. Looking forward, FJD could serve as valuable metric to ground future research, as it has the potential to help elucidate the most promising contributions within the scope of conditional generation.

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

## A   ILLUSTRATION OF FID AND FJD ON TWO DIMENSIONAL GAUSSIAN DATA

In this section, we illustrate the claim made in Section 1 that FID cannot capture intra-conditioning diversity when the joint distribution of two variables changes but the marginal distribution of one of them is not altered.

Consider two multivariate Gaussian distributions, $(X_1, Y_1) \sim \mathcal{N}(\mathbf{0}, \Sigma_1)$ and $(X_2, Y_2) \sim \mathcal{N}(\mathbf{0}, \Sigma_2)$, where

$$\Sigma_1 = \begin{bmatrix} 4 & 2 \\ 2 & 2 \end{bmatrix} \qquad \Sigma_2 = \begin{bmatrix} 2.1 & 2 \\ 2 & 2 \end{bmatrix}.$$

Figure 5 (left) shows $10,000$ samples drawn from each of these distributions, labeled as Dist1 and Dist2, respectively. While the joint distributions of $f_{X_1,Y_1}(X_1, Y_1)$ and $f_{X_2,Y_2}(X_2, Y_2)$ are different from each other, the marginal distributions $f_{Y_1}(Y_1)$ and $f_{Y_2}(Y_2)$ are the same ($Y_1 \sim \mathcal{N}(0, 2)$ and $Y_2 \sim \mathcal{N}(0, 2)$). Figure 5 (center) shows the histograms of the two marginal distributions computed from $10,000$ samples.

If we let $X_i$ take the role of the embedding of the conditioning variables (e.g., position) and $Y_i$ take the role of the embedding of the generated variables (i.e., images), then computing FID in this example would correspond to computing the FD between $f_{Y_1}$ and $f_{Y_2}$, which is *zero*. On the other hand, computing FJD would correspond to the FD between $f_{X_1,Y_1}$ and $f_{X_2,Y_2}$, which equals $0.678$. But note that Dist1 and Dist2 have different degrees of intra-conditioning diversity, as illustrated by Figure 5 (right), where two histograms of $f_{Y_i|X_i \in (0.9,1.1)}$ are displayed, showing marked differences to each other (similar plots can be constructed for other values of $X_i$). Therefore, this example illustrates a situation in which FID is unable to capture changes in intra-conditioning diversity, while FJD is able to do so.

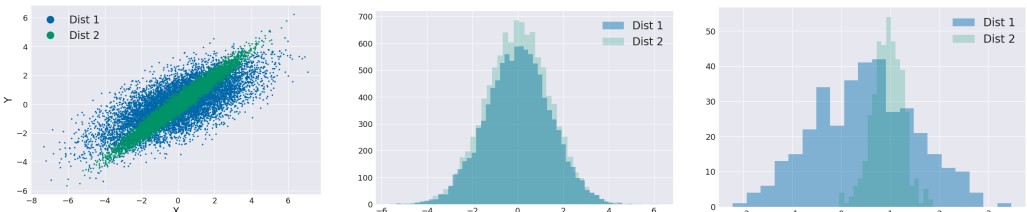

Figure 5: Left: samples from two multivariate Gaussian distributions. Center: Histograms of marginal distributions for the $Y$ variable. Right: Histogram of conditional distributions for $Y$ conditioned on $X \in (0.9, 1.1)$.

## B   EXPERIMENTAL SETTINGS FOR CALCULATING FJD

Important details pertaining to the computation of the FID and FJD metrics for different experiments included in this paper are reported in Table 5. For each dataset we report which conditioning modality was used, as well as the conditioning embedding function. Information about which split and image resolution are used for the reference and generated distributions is also included, as well as how many samples were generated per conditioning. Values for $\alpha$ reported here are calculated according to the balancing mechanism recommended in Section 4.2. Datasets splits marked by "-" indicate that the distribution is a randomly sampled subset of the full dataset.

## C   IMAGE QUALITY EVALUATION ON COCO-STUFF DATASET

We repeat the experiment initially conducted in Section 5.2 on a real world dataset to see how well FJD tracks image quality. Specifically, we use the COCO-Stuff dataset (Caesar et al., 2018), which provides class labels, bounding box annotations, and segmentation masks. We follow the same experimental procedure as outlined in Section 5.2: Gaussian noise is drawn from $\mathcal{N}(0, \sigma)$ and add to the images, where $\sigma \in [0, 0.25]$ and pixel values are normalized (and clipped after noise addition) to

Table 5: Settings used to calculate FJD in experiments.

| Dataset | Modality | Reference distribution | | Generated distribution | | | Embedding | α |
|---|---|---|---|---|---|---|---|---|
| | | Split | Img. res. | Split | Samp. per cond. | Img. res. | | |
| **dSprite** | Class | - | 64 | - | 1 | 64 | One-hot | 17.465029 |
| **dSprite** | BBox | - | 64 | - | 1 | 64 | AutoEncoder | 0.54181314 |
| **dSprite** | Mask | - | 64 | - | 1 | 64 | AutoEncoder | 0.38108996 |
| **ImageNet** | Class | Train | 128 | Valid. | 1 | 128 | One-hot | 17.810543 |
| **Facades** | Image | Train | 256 | Valid. + Test | 1 | 256 | InceptionV3 | 0.9376451 |
| **Maps** | Image | Train | 512 | Test | 1 | 512 | InceptionV3 | 1.1100407 |
| **Edges2Shoes** | Image | Train | 256 | Test | 1 | 256 | InceptionV3 | 0.73255646 |
| **Edges2Handbags** | Image | Train | 256 | Test | 1 | 256 | InceptionV3 | 0.7743437 |
| **CUB-200** | Text | Valid. | 256 | Valid. | 1 | 256 | Char-CNN-RNN | 4.2674055 |
| **COCO-Stuff** | Class | Valid. | 64 | Valid. | 1 | 64 | One-hot | 8.539345 |
| **COCO-Stuff** | BBox | Valid. | 64 | Valid. | 1 | 64 | AutoEncoder | 0.00351441 |
| **COCO-Stuff** | Mask | Valid. | 64 | Valid. | 1 | 64 | AutoEncoder | 0.001909862 |
| **COCO-Stuff** | Class | Valid. | 128 | Valid. | 1 | 128 | One-hot | 0.001909862 |
| **COCO-Stuff** | BBox | Valid. | 128 | Valid. | 1 | 128 | AutoEncoder | 0.0000399950188 |
| **COCO-Stuff** | Mask | Valid. | 128 | Valid. | 1 | 128 | AutoEncoder | 0.0150718 |

the range $[0, 1]$. The original dataset of clean images is used as the reference distribution, while noisy images are used to simulate a generated distribution with poor image quality. For the purposes of calculating FJD, we use N-hot encoding to embed the labels of the classes present in each image, and autoencoder representations for the bounding box and mask labels. As shown in Figure 6, FID and FJD both track image quality well, increasing as more noise is added to the generated image distribution.

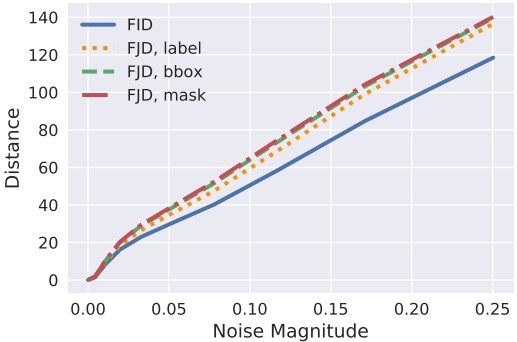

Figure 6: Comparison between FID and FJD for class, bounding box, and mask conditioning under varying noise levels for COCO-Stuff dataset. Evaluated at 128x128 resolution.

## D    CONDITIONAL CONSISTENCY EVALUATION WITH TEXT CONDITIONING

In order to test the effectiveness of FJD at detecting conditional inconsistencies in the text domain, we use the Caltech-UCSD Birds 200 dataset (Welinder et al., 2010). This dataset is a common benchmark for text conditioned image generation models, containing 200 fine-grained bird categories, 11,788 images, and 10 descriptive captions per images. Also included in the dataset are vectors of detailed binary annotations describing the attributes of the bird in each image. Each annotation indicates the presence or absence of specific features, such as `has\_bill\_shape::curved` or `has\_wing\_color::blue`.

Our goal in this experiment is to swap captions between images, and in this fashion introduce inconsistencies between images and their paired captions, while preserving the marginal distributions of images and labels. We compare attribute vectors belonging to each image using the Hamming distance to get an indication for how well the captions belonging to one image might describe another. Small Hamming distances indicate a good match between image and caption, while at larger values the captions appear to describe a very different bird than what is pictured (as demonstrated in Figure 7).

| | Caption | Hamming Distance |
|---|---|---|
| | This bird is fully covered in red except for some parts of wing and it has brown feet. | 0 |
| | A red bird with a short bill with a black cheek patch. | 13 |
| | This small bird is bright red with black wings, black eyeing, and short black beak. | 25 |
| | The small brown bird has a yellow beak and black round eyes. | 39 |
| | The body of the bird is ivory and the crown is bright red while the wing is black and ivory speckled. | 51 |

Figure 7: A summer tanager, as described by a variety of captions (ground truth caption highlighted in blue). The Hamming distance between attribute vectors associated with each caption and the ground truth caption provides an indication of how well each caption describes the image.

To test FJD we create two datsets: one which contains the original image-captions pairs from CUB-200 to act as the reference distribution, and another in which captions have been swapped to act as a generated distribution that has poor conditional consistency. Char-CNN-RNN embeddings are

used to encode the captions for the purposes of calculating FJD. In Figure 8 we observe that as the average Hamming distance across captions increases (i.e., the captions become worse at describing their associated images), FJD also increases. FID, which is unable to detect these inconsistencies, remains constant throughout.

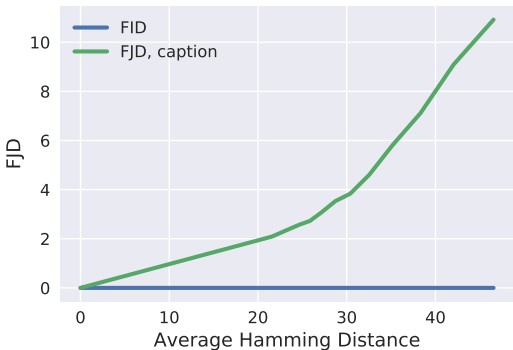

Figure 8: Change in FJD and FID with respect to the average Hamming distance between original captions and swapped captions. FJD increases as captions become worse at describing their associated image, while FID is insensitive.

## E    LIST OF SOURCES OF PRE-TRAINED MODEL

Table 6 includes the hyperlinks to all of the pretrained conditional generation models used in our experiments in Section 6.

Table 6: Source of pre-trained models evaluated in Section 6.

| Model | Source |
|---|---|
| SN-GAN | https://github.com/pfnet-research/sngan_projection |
| BigGAN | https://github.com/ajbrock/BigGAN-PyTorch |
| Pix2pix | https://github.com/junyanz/pytorch-CycleGAN-and-pix2pix |
| BicyleGAN | https://github.com/junyanz/BicycleGAN |
| MSGAN | https://github.com/HelenMao/MSGAN/ |
| MUNIT | https://github.com/nvlabs/MUNIT |
| HDGan | https://github.com/ypxie/HDGan |
| StackGAN++ | https://github.com/hanzhanggit/StackGAN-v2 |
| AttnGAN | https://github.com/taoxugit/AttnGAN |

## F    EFFECT OF $\alpha$ PARAMETER

The $\alpha$ parameter in the FJD equation acts as a weighting factor indicating the importance of the image component versus the conditional component. When $\alpha = 0$, then FJD is equal to FID, since we only care about the image component. As the value of $\alpha$ increases, the magnitude of the conditional component's contribution to the value of FJD increases as well. In our experiments, we attempt to find a neutral value for $\alpha$ that will balance the contribution from the conditional component and the image component. This balancing is done by finding the value of $\alpha$ that would result in equal magnitude between the image and conditioning embeddings (as measured by the average L2 norm of the embedding vectors).

Instead of reporting FJD at a single $\alpha$, an alternative approach is to calculate and plot FJD for a range of $\alpha$ values, as shown in Figure 9. Plotting $\alpha$ versus FJD allows us to observe any change in rank of models as the importance weighting on the conditional component is increased. Here we use the truncation trick to evaluate BigGAN (Brock et al., 2019) at several different truncation values $\sigma$. The

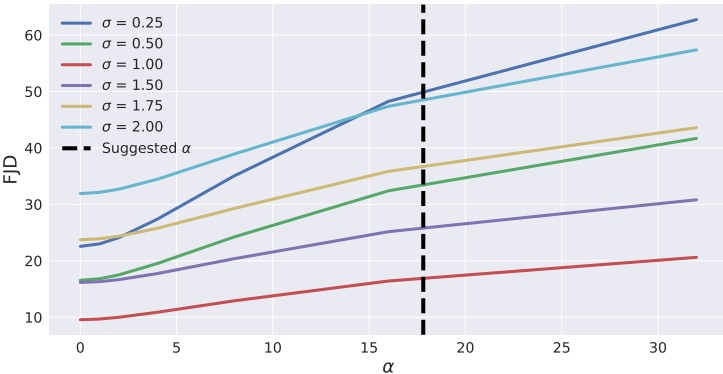

Figure 9: Alpha sweep for BigGAN at various truncation values $\sigma$. FID is equivalent to FJD at $\alpha = 0$. The black dashed line indicates the $\alpha$ value that is selected by calculating the ratio between the average $L_2$ norm of image and conditioning embeddings.

truncation trick is a technique wherein the noise vector used to condition a GAN is scaled by $\sigma$ in order to trade sample diversity for image quality and conditional consistency, without needing to retrain the model (as shown in Table 7).

Table 7: Comparison of BigGAN model evaluated with different truncation values $\sigma$. FJD is calculated at $\alpha = 17.8$. As $\sigma$ increases, classification accuracy decreases and diversity increases. Note that FID and FJD are consistent in their choice of the preferred model at $\sigma = 1.0$, however, the relative ranking of $\sigma = 0.25$ and $\sigma = 2.0$ is different between the two metrics.

| $\sigma$ | FJD $\downarrow$ | FID $\downarrow$ | Acc. $\uparrow$ | Diversity $\uparrow$ |
|---|---|---|---|---|
| 0.25 | 50.5 | 22.6 | **81.7** | 0.247 |
| 0.5 | 33.8 | 16.5 | 80.8 | 0.380 |
| 1.0 | **17.1** | **9.6** | 67.5 | 0.550 |
| 1.5 | 26.2 | 16.3 | 45.9 | 0.644 |
| 1.75 | 37.3 | 23.9 | 35.8 | 0.674 |
| 2.0 | 49.1 | 31.9 | 27.0 | **0.696** |

We find that in several cases, the ranking of models changes when comparing them at $\alpha = 0$ (equivalent to FID), versus comparing them using FJD at higher $\alpha$ values. Models with low truncation values $\sigma$ initially achieve good performance when $\alpha$ is also low. However, as $\alpha$ increases, these models rapidly drop in rank due to lack of sample diversity, and instead models with higher $\sigma$ values are favoured. This is most obvious when comparing $\sigma = 0.25$ and $\sigma = 1.75$ (blue and yellow lines in Figure 9) respectively.

## G    AUTOENCODER ARCHITECTURE

To create embeddings for the bounding box and mask conditionings evaluated in this paper we utilize a variant of the Regularized AutoEncoder with Spectral Normalization (RAE-SN) introduced by Ghosh et al. (2019) and enhance it with residual connections (Tables 8 and 9). For better reconstruction quality, we substitute the strided convolution and transposed convolution for average pooling and nearest neighbour upsampling, respectively. Spectral normalization (Miyato et al., 2018) is applied to all linear and convolution layers in the decoder, and an $L_2$ penalty is applied to the latent representation $z$ during training. Hyperparameters such as the weighting factor on the $L_2$ penalty and the number of dimensions in the latent space are selected based on which combination produces the best reconstructions on a held-out validation set.

In Tables 10 and 11 we depict the architecture for an autoencoder with $64 \times 64$ input resolution, but this can be scaled up or down by adding or removing residual blocks as required. $ch$ represents a channel multiplier which is used to control the capacity of the model. $M$ represents the number of

latent dimensions in the latent representation. $C$ indicates the number of classes in the bounding box or mask representation.

Table 8: ResBlock down

| Input $x$ |
| --- |
| $x \rightarrow \text{Conv}_{3 \times 3} \rightarrow \text{BN} \rightarrow \text{ReLU} \rightarrow out$ |
| $out \rightarrow \text{Conv}_{3 \times 3} \rightarrow \text{BN} \rightarrow \text{ReLU} \rightarrow out$ |
| $out \rightarrow \text{AvgPool}_{2 \times 2} \rightarrow out$ |
| $x \rightarrow \text{Conv}_{1 \times 1} \rightarrow res$ |
| $res \rightarrow \text{AvgPool}_{2 \times 2} \rightarrow res$ |
| $out + res \rightarrow \text{ReLU} \rightarrow out$ |

Table 9: ResBlock up

| Input $x$ |
| --- |
| $x \rightarrow \text{Conv}_{3 \times 3} \rightarrow \text{BN} \rightarrow \text{ReLU} \rightarrow out$ |
| $out \rightarrow \text{Conv}_{3 \times 3} \rightarrow \text{BN} \rightarrow out$ |
| $out \rightarrow \text{Upsample}_{2 \times 2} \rightarrow out$ |
| $x \rightarrow \text{Conv}_{1 \times 1} \rightarrow res$ |
| $res \rightarrow \text{Upsample}_{2 \times 2} \rightarrow res$ |
| $out + res \rightarrow \text{ReLU} \rightarrow out$ |

Table 10: Encoder

| Input $x \in \mathbb{R}^{64 \times 64 \times C}$ |
| --- |
| ResBlock down $C \rightarrow ch$ |
| ResBlock down $ch \rightarrow 2ch$ |
| ResBlock down $2ch \rightarrow 4ch$ |
| ResBlock down $4ch \rightarrow 8ch$ |
| Linear $8ch \times 4 \times 4 \rightarrow M$ |

Table 11: Decoder

| $z \in \mathbb{R}^{M}$ |
| --- |
| Linear $M \rightarrow ch \times 8 \times 8$ |
| BN $\rightarrow$ ReLU |
| ResBlock up $8ch \rightarrow 4ch$ |
| ResBlock up $4ch \rightarrow 2ch$ |
| ResBlock up $2ch \rightarrow ch$ |
| Conv $ch \rightarrow C$ |
| Tanh |

## H  FJD FOR MODEL SELECTION AND HYPERPARAMETER TUNING

In order to demonstrate the utility of FJD for the purposes of model selection and hyperparameter tuning, we consider the loss function of the generator from an auxiliary classifier GAN (ACGAN) (Odena et al., 2017), as shown in Equation 7 to 9. Here $S$ indicates the data source, and $C$ indicates the class label.

$$\mathcal{L}_S = E[\log P(S = real|X_{real}] + E[\log P(S = fake|X_{fake})] \tag{7}$$
$$\mathcal{L}_C = E[\log P(C = c|X_{real})] + E[\log P(C = c|X_{fake})] \tag{8}$$
$$\mathcal{L}_G = \lambda \mathcal{L}_C - \mathcal{L}_S \tag{9}$$

The generator loss $\mathcal{L}_G$ is maximized during training, and consists of two components: an adversarial component $\mathcal{L}_S$, which encourages generated samples to look like real samples, and a classification component $\mathcal{L}_C$, which encourages samples to look more like their target class. In this experiment we add a weighting parameter $\lambda$, which weights the importance of the conditional component of the generator loss. The original formulation of ACGAN is equivalent to always setting $\lambda = 1$, but it is unknown whether this is the most suitable setting as it is never formally tested. To this end, we train models on the MNIST dataset and perform a sweep over the $\lambda$ parameter in the range $[0, 5]$, training a single model for each $\lambda$ value tested. Each model is evaluated using FID, FJD, and classification accuracy to indicate conditional consistency. For FID and FJD we use the training set as the reference distribution, and generate $50,000$ samples for the generated distribution. Classification accuracy is measured using a pretrained LeNet classifier (LeCun et al., 1998), where the conditioning label is used as the groundtruth.

Scores from best performing models as indicated by FID, FJD, and classification accuracy are shown in Table 12. Sample sheets are provided in Figure 10, where each column is conditioned on a different digit from 0 to 9. We find that FID is optimized when $\lambda = 0.25$ (Figure 10a). This produces a model with good image quality, but almost no conditional consistency. Accuracy is optimized when $\lambda = 5.0$ (Figure 10c), yielding a model with good conditional consistency, but limited image quality. Finally, FJD is optimized when $\lambda = 1.0$ (Figure 10b), producing a model that demonstrates a balance between image quality and conditional consistency. These results demonstrate the importance of considering both image quality and conditional consistency simultaneously when performing hyperparameter tuning.

Table 12: Scores of ACGAN models trained with different values for conditioning weighting $\lambda$.

| $\lambda$ | FID ↓ | FJD ↓ | Accuracy ↑ |
|---|---|---|---|
| 0.25 | **56.87** | 79.65 | 12.22 |
| 1.0 | 62.39 | **65.31** | 74.90 |
| 5.0 | 115.23 | 119.10 | **98.01** |

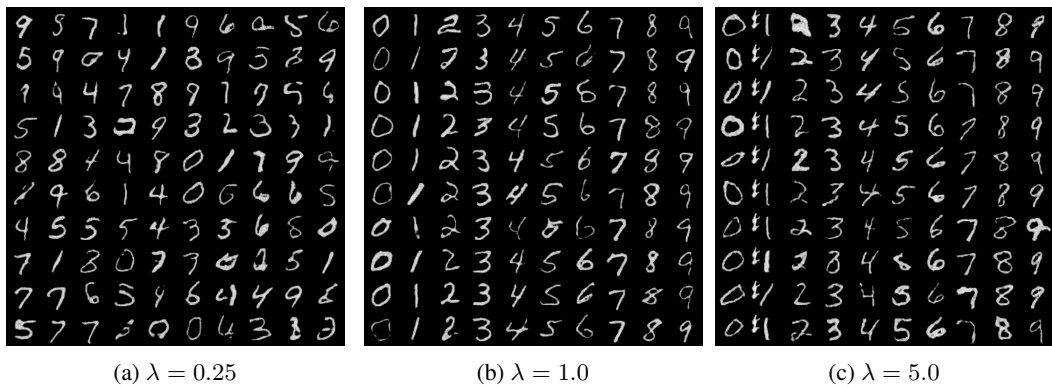

(a) $\lambda = 0.25$        (b) $\lambda = 1.0$        (c) $\lambda = 5.0$

Figure 10: Sample sheets for ACGAN model trained with different conditioning weighting $\lambda$. Each column is conditioned on a different digit, from 0 to 9. Low values of $\lambda$ produce models with very little conditional consistency, while overly large values of $\lambda$ yield models with reduced image quality and diversity.

## I   TRAINING AND EVALUATING WITH MULTI-LABEL, BOUNDING BOX, AND MASK CONDITIONING ON COCO-STUFF

To demonstrate FJD applied to multi-label, bounding box, and mask conditioning on a real world dataset, we train a GAN on the COCO-Stuff dataset (Caesar et al., 2018). To this end, we train three generative models, one for each conditioning type. Following (Johnson et al., 2018), we select only images containing between 3 and 8 objects, and also ignore any objects that occupy less than 2% of the total image area. Two image resolutions are considered: $64 \times 64$ and $128 \times 128$. We adopt a BigGAN-style model (Brock et al., 2019), but modify the design such that a single fixed architecture can be trained with any of the three conditioning types. See Section I.1 for architectural details. We train each model 5 times, with different random seeds, and report mean and standard deviation of both FID and FJD in Table 13. N-hot encoding is used as the embedding function for the multi-label conditioning, while autoencoder representations are used to calculate FJD for bounding box and mask conditioning.

In most cases we find that FID values are very close between conditioning types. A similar trend is observed in FJD at the $128 \times 128$ resolution. For models trained at $64 \times 64$ resolution however, we notice a more drastic change in FJD between conditioning types. Mask conditioning achieves the lowest FJD score, followed by multi-label conditioning and bounding box conditioning. This could indicate that the mask conditioning models are more conditionally consistent (or diverse) compared to other conditioning types.

Table 13: FJD / FID results averaged over 5 runs on COCO-stuff validation set with multi-label, bounding box (bbox) and mask conditionings for image resolutions $64 \times 64$ and $128 \times 128$.

|  | class conditioning | | bbox conditioning | | mask conditioning | |
|---|---|---|---|---|---|---|
|  | FJD ↓ | FID ↓ | FJD ↓ | FID ↓ | FJD ↓ | FID ↓ |
| **64** | $57.35 \pm 1.60$ | $40.75 \pm 1.38$ | $67.97 \pm 1.70$ | $41.81 \pm 1.50$ | $49.44 \pm 2.46$ | $41.27 \pm 2.36$ |
| **128** | $68.49 \pm 2.72$ | $50.74 \pm 2.31$ | $71.58 \pm 1.77$ | $51.78 \pm 1.55$ | $68.12 \pm 1.33$ | $46.02 \pm 1.22$ |

## I.1 COCO-STUFF GAN ARCHITECTURE

In order to modify BigGAN Brock et al. (2019) to work with multiple types of conditioning we make two major changes. The first change occurs in the generator, where we replace the conditional batch normalization layers with SPADE (Park et al., 2019). This substitution allows the generator to receive spatial conditioning such as bounding boxes or masks. In the case of class conditioning with a spatially tiled class vector, SPADE behaves similarly to conditional batch normalization. The second change we make is to the discriminator. The original BigGAN implementation utilizes a single projection layer (Miyato & Koyama, 2018) in order to provide class-conditional information to the discriminator. To extend this functionality to bounding box and mask conditioning, we add additional projection layers after each ResBlock in the discriminator. The input to each projection layer is a downsampled version of the conditioning that has been resized using nearest neighbour interpolation to match the spatial resolution of each layer. In this way we provide conditioning information at a range of resolutions, allowing the discriminator to use whichever is most useful for the type of conditioning it has received. Aside from these specified changes, and using smaller batch sizes, models are trained with the same hyperparameters and training scheme as specified in (Brock et al., 2019).

## I.2 SAMPLES OF GENERATED IMAGES

In this section, we present some random $128 \times 128$ samples of conditional generation for the models covered in Section I. In particular, Figures 11–13 show class, bounding box, and mask conditioning samples, respectively. Each row displays a depiction of conditioning, followed by 4 different samples, and finally the real image corresponding to the conditioning. As shown in Figure 11, conditioning on classes leads to variable samples w.r.t. object positions, scales and textures. As we increase the conditioning strength, we reduce the freedom of the generation and hence, in Figure 12, we observe how the variability starts appearing in more subtle regions. Similarly, in Figure 13, taking different samples per conditioning only changes the textures. Although the degrees of variability decrease as the conditioning strength increases, we obtain sharper, better looking images.

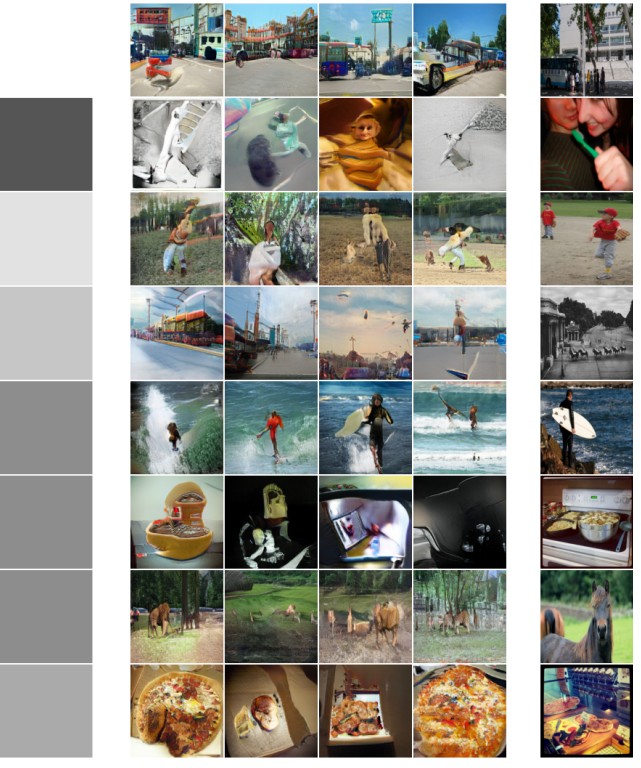

Figure 11: **Class-conditioning:** Conditioning, samples, and ground truth image for label-conditioned GAN. Greyscale intensity indicates class label.

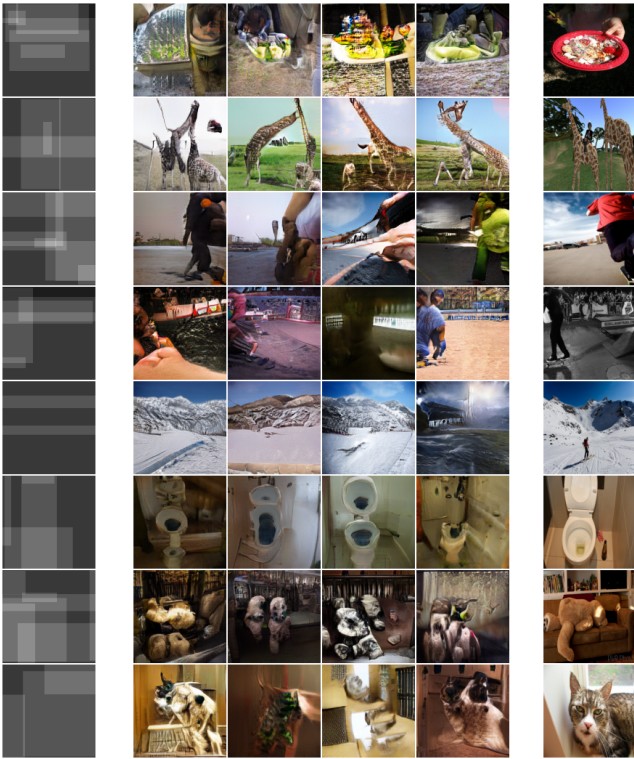

Figure 12: **Bounding box conditioning:** Conditioning, samples, and ground truth image for bounding box-conditioned GAN.

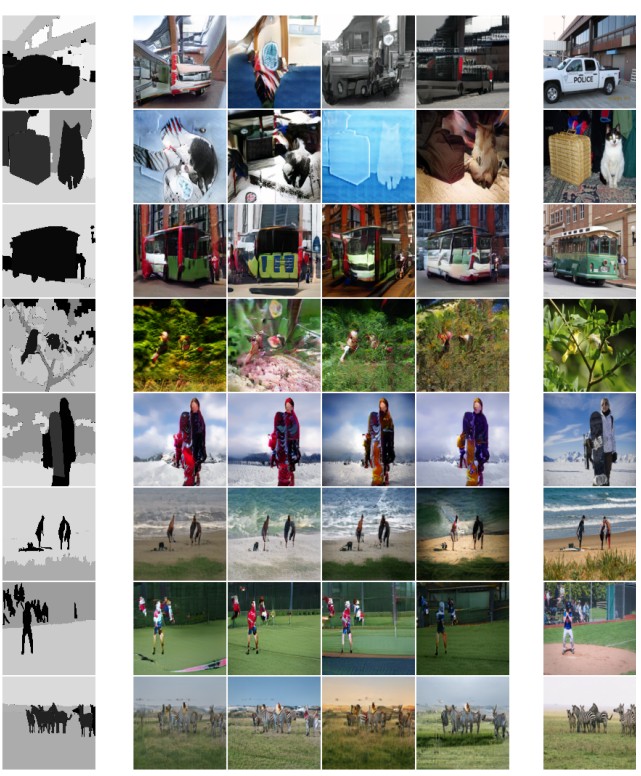

Figure 13: **Mask conditioning:** Conditioning, samples, and ground truth image for mask-conditioned GAN.

