# OpenReview forum: "On the Evaluation of Conditional GANs"
_ICLR.cc/2020/Conference — Reject_

### Official Review · AnonReviewer3 · 2019-10-11
**Official Blind Review #3**

**Rating:** 3

**Review:**

Summary:
This paper extends the Fréchet Inception distance (FID) to the conditional distribution. To this end, the authors use an additional embedding for the condition variables (class, image, text), and concatenate to the data embedding. The proposed metric, name the Fréchet joint distance (FJD), captures three desired properties of conditional generative models: sample quality, conditional consistency, and sample diversity. The authors demonstrate that the proposed metric indeed captures the properties using a synthetic (dSprite) dataset, and shows reasonable values for real datasets.

Pros:
- FJD is an intuitive extension of FID for conditional generative models.
- FJD can be applied to various types of conditions (e.g., image and text), which cannot be done by prior work (e.g., [1]).
- The paper is easy to read and experimental details are clearly stated.

Cons:

1. FJD is a straightforward extension of FID.

FJD simply follows the FID formula but concatenates the condition embedding to the original data embedding. It is a straightforward extension of FID and suffers from the design choice problems due to the concatenation, as stated below.

2. FJD requires many design choices and not theoretically justified.

As FJD requires an additional embedding function h, balancing parameter \alpha, and merging function g, it raises a burden of design choices. While the authors give some suggestions, they are not theoretically justified. Also, one may use the statistical distances [2] between data distribution p_data(x,c) and model distribution p_g(x,c) to evaluate conditional generative models in a principled way, e.g., measure the KL-divergence using the density ratios [3]. The advantage of FJD over such metrics is unclear, as stated below.

3. The advantage over the prior work is not clear.

FJD and FID show the same trend in all reported experiments (Table 2, 3, 4), hence the advantage of FJD is unclear. Also, one may measure the FID score on conditional distributions, i.e., \sum_c FID( p_data(x|c), p_g(x|c) ). It also captures the desired three properties and would be a strong baseline for FJD. Besides, while the authors aim to design a single metric to stand the models in a line, identifying the trade-offs of models may also be useful. For example, Improved PRD [4] provides the precision-recall trade-offs of generative models, which provides some insights for the models.


[1] Ravuri and Vinyals. Classification Accuracy Score for Conditional Generative Models. NeurIPS 2019.
[2] https://en.wikipedia.org/wiki/Statistical_distance
[3] Uehara et al. Generative Adversarial Nets from a Density Ratio Estimation Perspective. arXiv 2016.
[4] Kynkäänniemi et al. Improved Precision and Recall Metric for Assessing Generative Models. NeurIPS 2019.

**Experience Assessment:**

I have published one or two papers in this area.

**Review Assessment: Checking Correctness Of Derivations And Theory:**

I carefully checked the derivations and theory.

**Review Assessment: Checking Correctness Of Experiments:**

I carefully checked the experiments.

**Review Assessment: Thoroughness In Paper Reading:**

I read the paper thoroughly.

---

> ### Author Response · Authors · 2019-11-15
> **Response to Reviewer #3 (1 of 2)**
>
> We would like to thank the reviewer for the thorough analysis of our paper and the insightful comments. We have made several improvements to the paper based on your feedback, and have attempted to address your concerns below.
>
>
> Concern: FJD is a straightforward extension of FID.
>
> We think that the straightforward extension is beneficial, since many practitioners in the field are already familiar with FID, and, as the reviewer previously pointed out, the extension is intuitive. In fact, we see this more as a strength of our approach rather than a concern, and the reviewer seems to partially agree with this assessment (“FJD is an intuitive extension of FID for conditional generative models”). Moreover, to the best of our knowledge, we are the first to propose and demonstrate (in a series of carefully crafted and executed experiments) that such an extension to FD can actually fulfill the requirements of the evaluation of conditional image generation models and is a useful model selection technique.
>
>
> Concern: FJD requires many design choices and not theoretically justified.
>
> To be clear, FJD requires only three, not many, design choices beyond that of FID and other evaluation techniques (more details below). While we do not provide theoretical justification for our design choices, we did evaluate them empirically to confirm their suitability.
>
> Embedding function: All evaluation techniques require some sort of embedding function for dimensionality reduction, whether it is learned or pre-trained (consider the Inception score, FID, KID, Improved PR, CAS, etc.). This seems to be an inevitable requirement for evaluation metrics. In our preliminary experiments we found that FJD was fairly insensitive to the particular embedding function used, with different functions yielding similar trends.
>
> Alpha: In the paper we present a well-motivated heuristic for selecting alpha, which is based on the reasonable assumption that we desire a model which equally balances image quality and conditional consistency. As such, we present a simple scaling rule in which alpha is selected such that it balances the weight of the conditioning embedding with that of the image embedding. This has the added desirable effect of normalizing FJD values across embedding dimension and magnitude. We validated our design decision by generating multiple conditioning embedding spaces with varying dimensions and magnitudes. Without our scaling rule, each embedding space produces a different FJD value given the same input. However, when our scaling rule was applied, we observed that the FJD value was much more uniform across embeddings, which suggests that the scaling rule is performing as expected.
>
> Merging function: We evaluated several different options for the merging function, including random projection, addition, and multiplication (in the case that the image and conditioning embedding have the same number of dimensions). We measured the correlation between resulting FJD scores and conditional consistency, and found that concatenation consistently yielded the best correlation among the tested options.
>
>
> Concern: “Also, one may use the statistical distances [2] between data distribution p_data(x,c) and model distribution p_g(x,c) to evaluate conditional generative models in a principled way.”
>
> We agree. In fact, this is essentially what FJD is doing. Frechet distance (aka Wasserstein-2 distance), which we use to measure the distance between data and model distributions, is a statistical distance.
>
>
> Concern: FJD and FID show the same trend in all reported experiments (Table 2, 3, 4), hence the advantage of FJD is unclear.
>
> Please see our general comment above regarding similarity in ranking between FID and FJD.

---

> > ### Author Response · Authors · 2019-11-15
> > **Response to Reviewer #3 (2 of 2)**
> >
> > Concern: The advantage over the prior work is not clear.
> >
> > The method the reviewer indicates for measuring FID on conditional distributions was previously introduced as Intra-FID in [1] and is already discussed in our paper (see Related Work section - last paragraph). Certainly this method would also work in the right settings, but it has some limitations. Chief among these is that splitting the dataset into groups based on condition vastly reduces the number of samples used for each individual FID calculation. For example, consider a dataset with semantic mask conditioning. We can reasonably assume that no two masks in our dataset are identical. Therefore, each conditional distribution would contain only a single sample, making the computation of class-specific FID scores intractable. Even if we apply this to the most basic conditioning, such as class conditioning, we would still want at least 2048 samples per class in order to ensure that our covariance matrix is full rank (otherwise the matrix square root calculation might become unstable) [2]. This is a tall order, which some large scale datasets such as ImageNet don’t even meet (although to be fair there are ways to trade precision for improved computational stability in order to skirt this restriction).  A further challenge with Intra-FID is computation time, which scales linearly with the number of unique conditions. Given that FID can take upwards of 10 minutes to calculate depending on hardware constraints (according to [3]), calculating Intra-FID for a dataset such as ImageNet could potentially take days. In contrast, FJD has constant compute time with number of unique conditions.
> >
> > [1] cGANs with Projection Discriminator. ICLR 2018.
> > [2] https://github.com/bioinf-jku/TTUR
> > [3] https://github.com/ajbrock/BigGAN-PyTorch
> >
> >
> > Concern: Besides, while the authors aim to design a single metric to stand the models in a line, identifying the trade-offs of models may also be useful.
> >
> > Certainly, identifying trade-offs among models is useful, but not the goal of our current work. Our intention with FJD is to enable easy model selection and hyperparameter tuning of conditional GANs, which we achieve by summarizing performance by a single number. We have made changes to the abstract and introduction to clarify FJD’s intended use in model selection and cGAN benchmarking.
> >
> > Moreover, we would like to bring the reviewer’s attention the results that we report in Section 6 (as well as in appendix F). For all tested models, we report metrics that could already be used for trade-off analysis. In all experiments we report metrics that independently measure image quality, conditional consistency and conditional diversity.

---

### Official Review · AnonReviewer1 · 2019-10-22
**Official Blind Review #1**

**Rating:** 1

**Review:**

Summary
This paper mention that there are some critical drawbacks existing in IS (Inception Score) and FID (Fréchet Inception Distance) which are two popular metrics to measure image generation quality. However, IS and FID scores are initially designed for measuring unconditional distribution, which fails to capture the conditional consistency of conditional distribution. Thus, the authors propose to concatenate conditioned embedding h(y) with image feature vector f(x) to extend the FID metric. The authors also implement the method on a toy dataset to show the sensitivity of FJD on conditional consistency and several popular cGAN models to show the efficiency of FJD on real data.
Paper Strengths
  1. The method is intuitive and easy to implement.

Paper Weaknesses
1. Although this paper shows the problem of FID for capturing the conditional consistency sprightly with the toy dataset, however, this problem does not obviously show up on real data. Basically, FID can also give a good comparison of the different model as FJD

**Experience Assessment:**

I have published one or two papers in this area.

**Review Assessment: Checking Correctness Of Derivations And Theory:**

I did not assess the derivations or theory.

**Review Assessment: Checking Correctness Of Experiments:**

I assessed the sensibility of the experiments.

**Review Assessment: Thoroughness In Paper Reading:**

I made a quick assessment of this paper.

---

> ### Author Response · Authors · 2019-11-15
> **Response to Reviewer #1**
>
> Thank you for your review. Since your single concern was shared by the other reviewers, please see our general comment above, where we have addressed it.

---

### Official Review · AnonReviewer2 · 2019-10-29
**Official Blind Review #2**

**Rating:** 3

**Review:**

This paper proposes a variant of the use of Frechet Inception Distance (FID) for the evaluation and benchmarking of conditional GAN models. FID is a popular measure for comparing image distributions in the Inception v3 feature space, in terms of the means and variances of multivariate Gaussians fit to data samples from each distribution. The authors argue that FID is ill-suited for use with cGANs, in that they do not explicitly take into account conditional consistency or intra-conditioning diversity. The main contribution and basic idea of this paper is to create joint image-conditioning distributions from the image embedding and the conditioning embedding in the Inception embedding, and then to combine them (by default, through vector concatenation). These joint image-conditioning distributions are then fed to FID as per standard usage on image distributions alone. The authors refer to their approach as FJD (Frechet Joint Distance).

Although the authors propose FJD as a new technique, it should more properly be regarded as the direct use of FID on joint distributions. The main practical contribution of the paper thus reduces to the notion of concatenating the learned conditioning representation with the image representation. As a research contribution, this is in itself not very substantial. However, in their experimentation the authors do take care to show through examples the effect of their joint image-conditioning approach in assessing image quality, conditional consistency, and intra-conditioning diversity, under a variety of conditionings.

There are some issues that are not adequately addressed:

1) In all experimental cases FJD scores and FID scores correlate well, which undercuts the argument that FJD is superior to FID in assessing the performance of cGAN models. Can situations be experimentally demonstrated where that is not the case? In particular, what happens when the fit in image representation is dramatically better / worse than that of the conditioning representation? This situation is interesting, but not considered in this paper.

2) The effect of the dimensionality of the learned representations is not addressed. When concatenating vectors to produce a joint image-conditioning representation, the dimensionality increases, which would tend to produce larger FJD values than their corresponding FID values. The experimental results of this paper seem to confirm this. However, is
it really valid then to declare FJD as being somehow more sensitive to the conditioning simply by virtue of obtaining larger values and larger spreads than FID? It should be remembered that FID and FJD are *not* unitless measures.

Overall, in its current state the paper appears to be below the acceptance threshold.


**Experience Assessment:**

I have read many papers in this area.

**Review Assessment: Checking Correctness Of Derivations And Theory:**

I carefully checked the derivations and theory.

**Review Assessment: Checking Correctness Of Experiments:**

I carefully checked the experiments.

**Review Assessment: Thoroughness In Paper Reading:**

I read the paper thoroughly.

---

> ### Author Response · Authors · 2019-11-15
> **Response to Reviewer #2**
>
> We thank the reviewer for their careful reading of our paper and remarks. Below we address the concerns that were raised.
>
> Concern: the authors propose FJD as a new technique, it should more properly be regarded as the direct use of FID on joint distributions.
>
> We would like to clarify that in the paper we introduced FJD as a direct application of Fréchet distance (FD) to a joint embedding space (see Section 4). Similarly, FID is a direct application to FD to image embedding space obtained with an Inceptionv3 model (see Section 3). Moreover, when discussing our contribution we clearly state: (1) in the abstract - “In this paper, we propose the Fréchet Joint Distance (FJD), which is defined as the Fréchet distance between joint distributions of images and conditioning, allowing it to implicitly capture the aforementioned properties in a single metric”, and (2) in the introduction - “FJD computes the Fréchet distance on an embedding of the joint image-conditioning distribution, and introduces only small computational overhead over FID compared to alternative methods”.
>
>
> Concern: The main practical contribution of the paper thus reduces to the notion of concatenating the learned conditioning representation with the image representation. As a research contribution, this is in itself not very substantial.
>
> While our approach is very intuitive (aka simple), we see it rather as a strength of the approach and not limitation. Moreover, to the best of our knowledge, we are the first to propose and demonstrate (in a series of carefully crafted and executed experiments) that such extension to FD can actually fulfill the requirements of the evaluation of conditional image generation models and is a useful model selection technique. Thus, we argue that our contribution is novel and substantial.
>
>
> Concern: FJD scores and FID scores correlate well. Can situations be experimentally demonstrated where that is not the case?
>
> Please see our general comment above regarding similarity in ranking between FID and FJD.
>
>
> Question: what happens when the fit in image representation is dramatically better / worse than that of the conditioning representation?
>
> We were uncertain how to interpret this question. Is the reviewer referring to scenarios where samples have reduced image quality but good conditional consistency, and vice versa? Experiments we conducted in Section 5.2 and 5.3 cover both of these scenarios with synthetic examples. Our new experiment in Appendix H also demonstrates a real world example of GANs with good image quality and bad conditional consistency, as well as bad image quality and good conditional consistency. If the reviewer is referring to a relative “strength” of the image representation to the conditioning representation, please refer to Appendix F, where we study the behaviour of FJD as a function of different alpha parameter values. If this was not what you were referring to, please indicate so and we will do our best to clarify.
>
>
> Concern: the effect of the dimensionality of the learned representations is not addressed.  However, is it really valid then to declare FJD as being somehow more sensitive to the conditioning simply by virtue of obtaining larger values and larger spreads than FID?
>
> We would like to clarify that the only correct way to use FJD is to compare values for a fixed conditioning modality and fixed embedding function (thus the embedding dimensionality is constant). One should not compare numerical values between FID and FJD as well as for two FJDs obtained with different embedding functions or using different modalities. Thus, when discussing the results from Section 6, we draw our observations based on model rankings (within a single metric). However, we do recognize that the wording used in Sections 5.2 and 5.4 might indicate that it is correct to compare FJD to FID for a fixed model. We clarified it in the paper and corrected the discussion of the results in both Sections. We thank the reviewer for spotting this issue. Nevertheless, we would argue that by design FJD is sensitive to conditional inconsistencies while FID is blind to them (see the definitions of the metrics in Sections 3 and 4) and we prove this experimentally in Section 5 (e.g. see Figure 3) and in Appendix D.

---

### Author Response · Authors · 2019-11-15
**FID and FJD do not always give the same ranking of models**

We would like to thank all reviewers for their critical reviews and insights. A common concern among reviewers was that FID and FJD appeared to give similar model rankings and therefore the advantage of FJD over FID was unclear. We hope to address this common concern here.

R1: “Basically, FID can also give a good comparison of the different model as FJD”
R2: “In all experimental cases FJD scores and FID scores correlate well, which undercuts the argument that FJD is superior to FID in assessing the performance of cGAN models. Can situations be experimentally demonstrated where that is not the case?”
R3: “FJD and FID show the same trend in all reported experiments (Table 2, 3, 4), hence the advantage of FJD is unclear.”

In the paper, we demonstrated the differences in the behaviour of FID and FJD on the dSprite dataset (in the main body of the paper) as well as on COCO-Stuff (Appendix C) and the Caltech-UCSD Birds 200 dataset (Appendix D). We also showed in Table 7 in Appendix F that FJD can produce different rankings compared to FID when used for hyperparameter selection (in this case BigGAN results when varying the truncation parameter).  Thus, we argue that the properties captured by both metrics are very different.

Nevertheless, when it comes to the results reported in Section 6, it is true that both metrics produce consistent rankings and thus, FJD does not alter the FID-based model benchmark. However, this is not surprising since the tested models use very similar conditional consistency mechanisms, and therefore, the FJD scores are dominated instead by changes in image quality, leading FJD and FID to behave similarly as we mentioned in the original submission. We have modified Section 6 to clarify the interpretation of the reported results making sure not to create a false impression that the FJD-based ranking will always be consistent with the FID-based ranking.

To further demonstrate the differences between FID and FJD, we have added an additional experiment to Appendix H, in which we use both metrics to perform hyperparameter selection. In our experiment, we train a conventional auxiliary classifier GAN (ACGAN) on MNIST, with the addition of a weighting hyperparameter lambda on the classification portion of the generator loss. If lambda is set to 0, it is equivalent to training an unconditional GAN, since only the adversarial objective remains. As the value of lambda increases, the generator is further encouraged to produce samples that will be correctly classified by the classifier. For reference, the conventional ACGAN formulation is equivalent to lambda = 1.0. Our goal in this experiment is to determine the best value for lambda (i.e. one that balances image quality and conditional consistency), which we attempt to find by performing a hyperparameter sweep over lambda values. Models are evaluated using FID, FJD, and classification accuracy in order to demonstrate the differences in how each metric behaves. We hypothesize that FJD is a more useful metric in this setting because FID is restricted to measuring image quality and classification accuracy is restricted to conditional consistency, while FJD should consider both image quality and conditional consistency.

The results of this experiment support our hypothesis. We find that the model with lambda = 0.25 achieves the best FID score. This model has good image quality, but almost no conditional consistency. The model with lambda = 5.0 achieves the best classification accuracy which indicates that it has good conditional consistency, but it appears to be lacking in terms of image quality. Finally, the model with lambda = 1.0 is ranked as the best by FJD. This model achieves a good balance between image quality and conditional consistency. Generated samples from each model can be found in Appendix H.

While this experiment is elementary, it succinctly demonstrates that FID and FJD do not always yield the same ranking of models, and that FID alone is not enough to gauge the performance of conditional GANs.

---

### Decision · Program_Chairs · 2019-12-19

**Decision:**

Reject

**Comment:**

The paper presents an extension of FID for conditional generation settings. While it's an important problem to address, the reviewers were concerned about the novelty and advantage of the proposed method over the existing methods. The evaluation is reported on toy datasets, and the significance is limited.